

# Differences in stability of seed-associated microbial assemblages in response to invasion by phytopathogenic microorganisms

Samir Rezki[1], Claire Campion[2], Beatrice Iacomi-Vasilescu[3], Anne Preveaux[1], Youness Toualbia[2], Sophie Bonneau[1], Martial Briand[1], Emmanuelle Laurent[4], Gilles Hunault[5], Philippe Simoneau[2], Marie-Agnès Jacques[1] and Matthieu Barret[1]

[1] Institut de Recherche en Horticulture et Semences, Institut National de la Recherche Agronomique, Beaucouzé, France
[2] Institut de Recherche en Horticulture et Semences, Université d'Angers, Beaucouzé, France
[3] Universitatea de Științe Agronomice și Medicină Veterinară din București, Bucharest, Romania
[4] Federation Nationale des Agriculteurs Multiplicateurs de Semences, Brain sur l'Authion, France
[5] Laboratoire d'Hémodynamique, Interaction Fibrose et Invasivité tumorale Hépatique, Université d'Angers, Angers, France

Corresponding author
Matthieu Barret,
matthieu.barret@angers.inra.fr

## ABSTRACT

Seeds are involved in the vertical transmission of microorganisms from one plant generation to another and consequently act as reservoirs for the plant microbiota. However, little is known about the structure of seed-associated microbial assemblages and the regulators of assemblage structure. In this work, we have assessed the response of seed-associated microbial assemblages of *Raphanus sativus* to invading phytopathogenic agents, the bacterial strain *Xanthomonas campestris* pv. *campestris* (*Xcc*) 8004 and the fungal strain *Alternaria brassicicola* Abra43. According to the indicators of bacterial (16S rRNA gene and *gyrB* sequences) and fungal (ITS1) diversity employed in this study, seed transmission of the bacterial strain *Xcc* 8004 did not change the overall composition of resident microbial assemblages. In contrast seed transmission of Abra43 strongly modified the richness and structure of fungal assemblages without affecting bacterial assemblages. The sensitivity of seed-associated fungal assemblage to Abra43 is mostly related to changes in relative abundance of closely related fungal species that belong to the *Alternaria* genus. Variation in stability of the seed microbiota in response to *Xcc* and Abra43 invasions could be explained by differences in seed transmission pathways employed by these micro-organisms, which ultimately results in divergence in spatio-temporal colonization of the seed habitat.

## INTRODUCTION

Seeds are not only carriers of plants genetic information but are also involved in the vertical transmission of microorganisms from one plant generation to another and consequently act as reservoirs for the plant microbiota (*Baker & Smith, 1966*; *Nelson, 2004*). The activity of

seed-associated microbial assemblages is significant for plant growth and plant health since these microbial assemblages may release seed dormancy through production of cytokinins (*Goggin et al., 2015*) or limit the installation of microbial invader (*Bacilio-Jimenez et al., 2001*). Although transmission of microorganisms from plant to seed is the primary source of inoculum for the plant, relatively little is known about the structure of seed-associated microbial assemblages and the regulators of assemblage structure (*Johnston-Monje & Raizada, 2011*; *Van Overbeek et al., 2011*; *Lopez-Velasco et al., 2013*; *Links et al., 2014*; *Barret et al., 2015*; *Klaedtke et al., 2015*).

Seeds acquire their microbiome by three major pathways: (i) internal transmission through the vascular system, (ii) floral transmission by the stigma and (iii) external transmission via contact of the seed with microorganisms present on fruits, flowers or residues (*Maude, 1996*). According to the transmission pathway, seed-borne microorganisms can therefore be located on different micro-habitats ranging from the testa to the embryo (*Singh & Mathure, 2004*; *Dutta et al., 2012*; *Tancos et al., 2013*). While the internal transmission by the host xylem is restricted to vascular pathogens or endophytic micro-organisms (*Maude, 1996*), many plant associated micro-organisms are potentially transmitted to the seed by the floral pathway (*Shade, McManus & Handelsman, 2013*; *Aleklett, Hart & Shade, 2014*). Indeed, the floral pathway might allow the transmission of biocontrol microorganisms (*Spinelli et al., 2005*) and phytopathogens (*Darsonval et al., 2008*; *Darrasse et al., 2010*; *Terrasson et al., 2015*). Finally, the external pathway is probably the most permissive way of microorganism transmission from plant to seed, although very little data are currently available in the literature (*Ngugi & Scherm, 2006*).

Owing to the importance of seed transmission in emergence of diseases in new planting areas, the processes involved in the transmission of microorganisms from plant to seed have been mainly documented for phytopathogenic agents. The molecular determinants involved in successful transmission of microorganisms from plant to seed have been notably studied in bacteria related to the *Xanthomonas* genus (*Darsonval et al., 2008*; *Darsonval et al., 2009*; *Darrasse et al., 2010*; *Dutta et al., 2014*) and fungi that belonged to *Alternaria brassicicola* (*Pochon et al., 2012*; *Pochon et al., 2013*). Key molecular determinants like the bacterial Type III Secretion System (*Darsonval et al., 2008*), bacterial adhesins (*Darsonval et al., 2009*), fungal class III histidine-kinase or dehydrin like proteins (*Pochon et al., 2013*) have been shown to be involved in seed transmission. In turn, seeds may respond to pathogen transmission through activation of plant defenses and subsequent repression of seed maturation pathways (*Terrasson et al., 2015*).

Although the host immune system is a decisive environmental filter to limit the installation of an invader (*Jones & Dangl, 2006*), the host-associated microbial community may also strongly prevent this invasion (*Mendes et al., 2011*). The resistance of microbial community to invasion is linked to its level of diversity, since highly diverse microbial community are usually less sensitive to invasion (*Jousset et al., 2011*; *Van Elsas et al., 2012*) as a result of enhanced competition for resources within species-rich community (*Mallon, Elsas & Salles, 2015*; *Mallon et al., 2015*; *Wei et al., 2015*). Because of its relative low microbial diversity (*Lopez-Velasco et al., 2013*; *Links et al., 2014*; *Barret et al., 2015*; *Klaedtke et al.,*

*2015*) compared to other plant habitats such as the phyllosphere (*Vorholt, 2012*) or the rhizosphere (*Hacquard et al., 2015*), mature seed is an interesting experimental model to study biological disturbance. Moreover, mature seeds have low moisture content and are almost metabolically inactive (*Dekkers et al., 2015*) which suggests that associated micro-organisms are probably dormant and that the structure of microbial assemblages is in a stable state. Hence shifts in assemblage structure are likely to reflect the outcome of the response of the microbial assemblage during seed transmission of phytopathogenic micro-organisms.

The aim of the present work was to analyze the impact of two microbial invaders, namely the bacterial strain *Xanthomonas campestris* pv. *campestris* (*Xcc*) 8004 and the fungal strain *A. brassicicola* (*Ab*) Abra43 on the genetic structure of microbial assemblages associated to seeds of *Raphanus sativus*. We choose these two microbial invaders since they differ in their seed transmission pathways. Indeed *Xcc* is mostly transmitted from plant to seeds by the systemic and floral pathways (*Cook, Larson & Walker, 1952*; *Van Der Wolf & Van Der Zouwen, 2010*; *Van der Wolf, Van der Zouwen & Van der Heijden, 2013*), while *Ab* is transmitted by the external pathway and thus mostly restricted to the testa (*Knox-Davies, 1979*). Radish (*Raphanus sativus*) was used as an experimental system since this plant has a high seed multiplication ratio (1:500 on averages) and a low microbial diversity compared to other Brassicaceae species (*Barret et al., 2015*). Profiling of seed-associated microbial assemblages was performed on mature seeds harvested from uninoculated plants or plants inoculated either with Abra43 or *Xcc* 8004 through sequencing of two bacterial taxonomic markers (16S rRNA gene and *gyrB*) and one fungal taxonomic marker (ITS1 region of the fungal internal transcribed spacer). This work revealed that the plant pathogenic fungal strain Abra43 had a significant impact on the fungal assemblages, while *Xcc* 8004 transmission did not impact the structure of seed microbiota.

## MATERIALS & METHODS

### Site description, inoculation process and seed collection

Experiments were performed at the FNAMS experimental station (47°28′12.42″N, 0°23′44.30″W, Brain-sur-l'Authion, France) in 2013 and 2014 on 3 distinct plots (5 × 10 m). Each plot was initially sown with *Raphanus sativus* var. Flamboyant5 on March 28th 2013 with a commercial seed lot. According to microbiological and community profiling analyses this seed sample was neither contaminated with *Xanthomonas campestris* pv. *campestris* (*Xcc*) nor *Alternaria brassicicola* (*Barret et al., 2015*). One plot (X2013) was spray-inoculated (approximately 100 ml m$^{-2}$) at floral stage (June 18th 2013) with *Xcc* strain 8004. *Xcc* 8004 is a spontaneous rifampicin-resistance strain derived from *Xcc* NCPPB 1145 (*Qian et al., 2005*). The bacterial strain *Xcc* 8004 was cultivated on Tryptic Soy Agar (TSA) 100% (17 g l$^{-1}$ tryptone, 3 g l$^{-1}$ soybean peptone, 2.5 g l$^{-1}$ glucose, 5 g l$^{-1}$ NaCl, 5 g l$^{-1}$ K2HPO4, and 15 g l$^{-1}$ agar) medium with rifampicin 50 mg l$^{-1}$ for 2 days at 28 °C. Bacterial colonies of *Xcc* 8004 were suspended in sterile deionized water at a final concentration of 1·10$^7$ CFU/ml. Another plot (A2013) was inoculated with *Alternaria brassicicola* strain Abra43 (*Avenot et al., 2005*) at the end of flowering (July 05th 2013) and at silique-formation stage (July 17th 2013) following protocol described earlier
(*Iacomi-Vasilescu et al., 2008*). The last plot (C2013) was not inoculated and subsequently used as a control plot. Plots were watered 4 h before inoculation in order to have a high relative humidity (approximately 80%).

A second experimentation was performed in 2014. Seeds harvested from C2013 were sown on April 2th, 2014 following the same experimental design as in 2013. While *Alternaria brassicicola* was not detected within C2013 sample, a residual contamination of *Xcc* was observed (Figs. 1A and 1C, see results section for further details). The only difference between 2013 and 2014 experiments rely in *Xcc* 8004 inoculations. While in 2013 plants were inoculated once, two inoculations were performed in 2014 in order to increase the efficiency of *Xcc* 8004 transmission from plant to seed. Inoculation of *Xcc* 8004 were done at the beginning (June 18th 2014) and at the end (June 25th 2014) of flowering stage. Inoculation with *Ab* (June 26th 2014 and July 7th 2014) was performed in one plot (A2014) using the same protocol as during the 2013 experimentation. The last plot (C2014) was left uninoculated. We did not observed any leaf spot symptoms on plants inoculated with *Xcc* 8004. In contrast, some dark spots on pods were observed on plants inoculated with *Alternaria brassicicola* Abra43.

At the mature seeds stage, seeds from plots C2013, A2013 and X2013 were harvested on September, 5th 2013, while seeds from plots C2014, A2014 and X2014 were collected on August, 22th 2014. Eighteen plants of each plot were collected and seeds were harvested manually for each individual plant. The remaining plants of each plot were harvested with a threshing machine leading to 6 seed lots (C2013, C2014, A2013, A2014, X2013 and X2014). Each of these seed lots was further divided into subsamples of 1,000 seeds (as assessed by 1,000 seeds weight). Seeds were conserved between one to two months at 9 °C and 50% relative humidity prior to DNA extraction.

## Culture-based detection of phytopathogenic agents on seed samples

The transmission of *Xcc* 8004 from plants to seeds was initially evaluated through microbiological analysis of seed lots harvested with the threshing machine. Individual seeds were deposited in a 96 well-plate and soaked in 200 µL of phosphate buffer saline (PBS) supplemented with 0.05% ($v/v$) of Tween® 20 during 2 h and 30 min at room temperature under constant agitation (140 rpm). Then 10 µl of each well was spread on TSA 10% medium supplemented with rifampicin (50 mg/l). Plates were incubated at 28 °C for 2 days and the presence of *Xcc* 8004 was evaluated for each individual seed. Contamination rate of each seed lot is a mean of six biological replicates, each containing 96 seeds. The transmission of *Alternaria brassicicola* strain Abra43 (*Ab*) from plant to seeds was assessed by standard plating technique (*Iacomi-Vasilescu et al., 2008*). Briefly, 10 individual seeds were deposited on potato-dextrose agar (4 g l$^{-1}$ potato extract, 20 g l$^{-1}$ dextrose, 15 g l$^{-1}$ agar) and plates were incubated at 25 °C for 5 days. *Alternaria brassicicola* isolates were further purified by monospore isolation. Contamination rate of each seed lot is a mean of 6 biological replicates, each biological replicate containing 100 seeds (10 plates per seed lot).

## DNA extraction

DNA extraction were performed according to standard procedures recommended by the International Seed Testing Association (ISTA, https://www.seedtest.org/en/home.html).

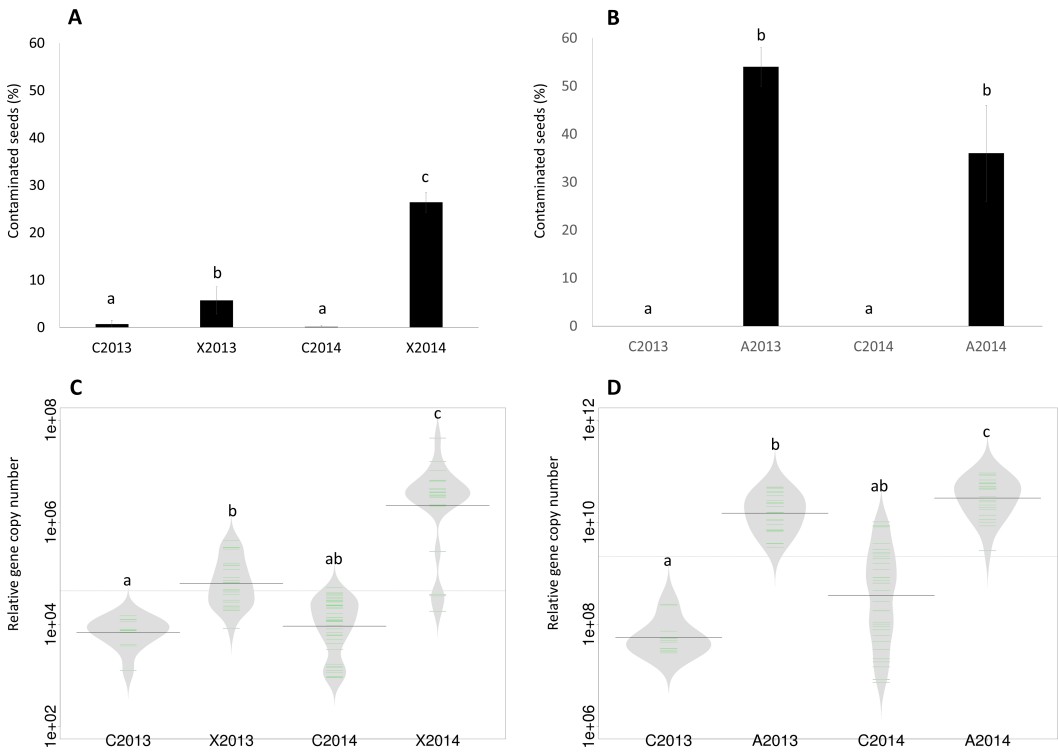

**Figure 1** **Assessment of seed contamination by *X. campestris* pv. *campestris* 8004 and *A. brassicicola* Abra43.** The contamination rate of seed samples by *X. campestris* pv. *campestris* 8004 (A) and *A. brassicicola* Abra43 (B) was assessed through microbiological analysis of 96 and 100 individual seeds, respectively. Contamination rates are the mean of 6 independent biological replicates performed for each experimental year. Quantitative detection of *Xcc* (C) was performed on seed samples harvested from uninoculated plants (C2013 and C2014) and plants inoculated with *Xcc* (X2013 and X2014) through qPCR with primers targeting XC_1533. Quantitative detection of *Ab* (D) was performed on seed samples harvested from uninoculated plants (C2013 and C2014) and plants inoculated with *Ab* (A2013 and A2014) through qPCR with primers and probe targeting *AbDhn1*. The green lines represent the number of each target gene in the different samples, while bold dark black lines represent the median. The grey area indicates the density of distribution. Differences in contamination rate and relative abundance were considered significant at a $p$-value $\leq 0.01$ (as assessed by 2-sample test for equality of proportions and ANOVA with post hoc Tukey's HSD test, respectively).

Briefly, seeds harvested manually on each plant or 1,000 seeds subsamples harvested mechanically were transferred in sterile tubes containing 25 mL of PBS supplemented with 0.05% ($v/v$) of Tween® 20. Samples were incubated during 2 h and 30 min at room temperature under constant agitation (140 rpm). Seed soaking procedures are routinely used to assess the presence of seed-borne pathogens within seed tissues (*Gitaitis & Walcott, 2007*). Indeed, microorganisms located within the seed coat as well as in the funiculus are released in the suspension with such experimental procedure. Suspensions were centrifuged (6,000 $\times$ $g$, 10 min, 4 °C) and pellets were resuspended in approximately 2 ml of supernatant and transferred in microtubes. Total DNA was extracted with the Power Soil DNA Kit (MoBio Laboratories) from 148 seed suspensions following procedure described earlier (*Barret et al., 2015*).

## Molecular detection of phytopathogenic agents on seed samples

Seed transmission of phytopathogenic agents was also monitored by quantitative PCR. The number of copies of the predicted gene XC_1533 encoding a hypothetical protein was used as an estimator of the number of Xcc 8004 cells per seed samples (*Rijlaarsdam et al., 2004*). A portion of this gene was amplified with the primer set Zup4F/Zup4R (Table S6). Data normalization between seed samples was performed with a portion of the 16S rRNA gene, using the primers 926F/1062R (Table S1). All reactions were performed in 25 µl qPCR reaction using 12.5 µl of SYBR Green Master Mix (MESA BLUE qPCR MasterMix Plus for SYBR Assay; Eurogentec, Cologne, Germany), 2 µl of DNA (10 ng µl$^{-1}$) and 0.5 µL of each primer (10 µM). Amplification conditions were 5 min at 95 °C, followed by 40 two-step cycles of 95 °C (15s) and 60 °C (60s).

The number of copies of the gene AbDhn1 encoding a dehydrin-like protein was used as an estimator of the number of Abra43 cells (*Pochon et al., 2013*) using the primer set AbraDHN1-Tq-F/AbraDHN1-Tq-R and a TaqMan MGB probe (Table S1). Reactions were conducted using StepOnePlus$^{TM}$qPCR system (Applied Biosystems). qPCR reaction were performed in 20 µl using 10 µl of Master Mix, 2 µl of TaqMan probe (250 nM), 2 µl of each primer (0.3 µM and 0.9 µM for the forward and the reverse primer respectively) and 2 µl of DNA (10 ng µl$^{-1}$). Amplification conditions were 10 min at 95 °C, followed by 40 two-step cycles of 95 °C (15s) and 60 °C (60s). Data normalization between seed samples was performed with a portion of the $\beta$-actin gene, using the primers ACT 512-F/ACT 783-R (Table S6) and amplification conditions described previously (*Carbone & Kohn, 1999*).

## Assessment of microbial diversity

PCR amplification was conducted on 148 DNA samples and 2 artificial microbial community samples containing a mixture of 15 DNA extracted from different bacterial strains (*Barret et al., 2015*). The 148 environmental DNA samples correponded to seed samples harvested from individual plants (18 plants per plot) or with a threshing machine for each plot (6 subsamples on average for each plot). Amplification were performed with the primer sets 515F/806R, gyrB_aF64/gyrB_aR353 and ITS1F/ITS2 (Table S1) following procedures described earlier (*Barret et al., 2015*). Amplicon libraries were mixed with 7.5% PhiX control according to Illumina's protocols. A total of four sequencing runs were performed for this study with MiSeq Reagent Kits v2 (500 cycles).

Sequence analyses were performed with Mothur v1.31.2 (*Schloss et al., 2009*) using standard operating procedure (*Kozich et al., 2013*) described earlier (*Barret et al., 2015*). Briefly,16S rRNA gene and *gyrB* sequences were aligned against the 16S rRNA gene SILVA alignment and a *gyrB* reference alignment, respectively. Chimeric sequences were detected with UCHIME (*Edgar et al., 2011*) and subsequently removed from the dataset. Moreover, *gyrB* sequences containing stop codon were discarded. Taxonomic affiliation of 16S rRNA gene and *gyrB* sequences was performed with a Bayesian classifier (*Wang et al., 2007*) (80% bootstrap confidence score) against the 16S rRNA gene training set (v9) of the Ribosomal Database Project (*Cole et al., 2009*) or against an in-house *gyrB* database created with sequences retrieved from the IMG database (*Markowitz et al., 2012*; *Barret et al., 2015*). Unclassified sequences (0.001% of the 16S rRNA gene sequences) or sequences belonging

to Archaea (0.002%), chloroplasts (0.9%) or mitochondria (0.004%) were discarded. Sequences were divided into groups according to their taxonomic rank (level of order) and then assigned to operational taxonomic units (OTUs) at 97% identity cutoff for 16S rRNA gene and 98% identity for *gyrB* sequences. The variable ITS1 regions of ITS sequences were extracted with the Perl-based software ITSx (*Bengtsson-Palme et al., 2013*). Then sequences were clustered at a 97% identity cut-off using Uclust (*Edgar, 2010*) and taxonomic affiliation was performed with a Bayesian classifier (*Wang et al., 2007*) (80% bootstrap confidence score) against the UNITE database (*Abarenkov et al., 2010*). To improve the resolution of the taxonomic classification of ITS1 sequences, we performed a reciprocal blast analysis at a minimum cut-off of 97% with representatives OTUs sequences and available ITS1 sequences of *Alternaria* type strains (*Woudenberg et al., 2013*). In order to enhance the reproducibility of community profiles, abundant OTUs (aOTU) representing at least 0.1% of the library size were used for microbial community analyses (*Barret et al., 2015*).

In order to avoid biases introduced by unequal sampling, total counts were divided by library size of each sample sequenced (*McMurdie & Holmes, 2014*). Both alpha and beta diversity indexes were calculated with Mothur (*Schloss et al., 2009*). Richness was defined as the number of different OTUs and aOTUs per sample. Hierarchical clustering of different seed samples was performed using an average linkage method based on Bray–Curtis dissimilarity index (*Bray & Curtis, 1957*) and on unweighted UniFrac distances (*Lozupone & Knight, 2005*). Analysis of similarity (ANOSIM) was used to assess the effects of the different conditions on the microbial community structure. Moreover, canonical analysis of principal coordinates (CAP) was conducted to measure the relative influence of (i) the phytopathogenic agent, (ii) the harvesting year and (iii) the harvesting method on microbial $\beta$-diversity. CAP analyses were performed with the function capscale of vegan.

Correlation between aOTUs were calculated with Sparse Correlation for Compositional data algorithm (*Friedman & Alm, 2012*) implemented in Mothur. Statistical significance of the inferred correlations was assessed with a bootstrap procedure (100 replications). Only correlations with pseudo $p$-value $\leq 0.001$ were represented in the network using the R package qgraph (*Epskamp et al., 2012*). Changes in relative abundance of aOTUs between the different experimental conditions (C, X and A) were assessed with LEfSE (*Segata et al., 2011*). aOTUs were defined as significantly enriched or depleted in one treatment at a $P$ value $\leq 0.05$ and a LDA score $>2$.

All sequences have been deposited in the ENA database under the accession number PRJEB9588.

## RESULTS

The impact of pathogen transmission on the structure of seed-associated microbial assemblages was assessed on radish seed lots harvested in 2013 and 2014 from plots inoculated with the bacterial strain *Xanthomonas campestris* pv. *campestris* 8004 (X2013 and X2014 plots), with the fungal strain *Alternaria brassicicola* Abra43 (A2013 and A2014 plot) and from uninoculated plants (C2013 and C2014 for control plots).

## Efficient transmission of phytopathogenic microorganisms to seeds

The seed transmission of *Xcc* 8004 and *Ab* Abra43 was first evaluated by standard microbiological approaches on seed samples collected in 2013 and 2014. According to these microbiological analyses, *Ab* was not detected on seeds harvested from control plots (C2013 and C2014), while a residual *Xcc* contamination of 0.69% and 0.13% was observed in seeds from C2013 and C2014 samples, respectively (Figs. 1A and 1B). However, the incidence of *Xcc* increased significantly in X2013 and X2014 samples ($P < 0.01$, as assessed by 2-sample test for equality of proportions) with 6% and 26% of seeds contaminated, respectively (Fig. 1A). *Ab* was not detected in control samples, nevertheless a significant increase of *Ab* incidence ($P < 0.01$) was observed in A2013 and A2014 samples with 54% and 36% of seed contaminated (Fig. 1B), respectively.

To confirm these results, qPCR experiments were performed on DNA extracted from seed samples with primers and probes targeting XC_1533, a single-copy gene of *Xcc* encoding a hypothetical protein (*Rijlaarsdam et al., 2004*), and *AbDhn1*, a single-copy gene of *Ab* encoding a dehydrin-like protein (*Pochon et al., 2013*). A significant increase ($P < 0.01$, as assessed by ANOVA with post hoc Tukey's HSD test) in copy number of XC_1533 was observed on X2013 ($1 \cdot 10^5$ copies) and X2014 ($4 \cdot 10^6$ copies) samples in comparison to control samples (Fig. 1C). Similarly, the number of copy of *AbDhn1* also increased ($P < 0.01$) in A2013 ($1.5 \cdot 10^{10}$) and A2014 ($4.5 \cdot 10^{10}$) (Fig. 1D). These differences were not due to variation in DNA amounts between seed samples since the copy numbers of 16S rRNA and $\beta$-actin genes were not significantly different between seed samples (Fig. S1). Altogether microbiological and qPCR analyses highlighted an effective transmission of *Xcc* 8004 and *Ab* Abra43 from plant to seed.

## Seed-associated fungal assemblages are impacted by *Ab* Abra43 transmission

The structure of 148 microbial assemblages associated to seeds harvested in C2013, C2014, A2013, A2014, X2013 and X2014 was assessed through amplification and subsequent sequencing of two bacterial molecular markers (16S rRNA gene and *gyrB*) and one fungal molecular marker (ITS1). A total of 7,870,622 (16S rRNA gene), 24,355,191 (*gyrB*) and 8,799,598 (ITS1) paired-end reads were obtained (Table S2). Reads were assembled in quality sequences and grouped into operational taxonomic units (OTUs) at ≥97% sequence identity for 16S rRNA gene and ITS1 sequences and ≥98% sequence identity for *gyrB* (Table S2). To increase the reproductibility of OTU detection between samples, OTUs with a relative abundance ≥0.1% of the library size were defined as abundant OTUs (*Barret et al., 2015*). However, this threshold remove rare OTUs, which contributes to a large amount of diversity observed within microbial assemblages (*Shade et al., 2014*). Therefore, subsequent analyses were performed (i) on every OTUs including abundant and rare OTUs and (ii) on abundant OTUs (aOTUs) only. Since seed samples used in this study have been either harvested manually on individual plants or mechanically with a threshing machine (see experimental procedures for further informations), the influence of the harvesting method on the structure of seed-associated microbial assemblages was first investigated. According to ANOSIM tests, the structure of seed-associated microbial assemblages on seed samples harvested manually are not significantly different from samples collected with

**Table 1  Analysis of similarity of seed-associated microbial assemblages.** Analysis of similarity (ANOSIM) was used to assess the robustness of the hierarchical clustering analyses (Bray–Curtis dissimilarity measure and unweighted Unifrac distance). *P*-values are displayed in each column. Only *P*-values highlighted in bold are considered as significant.

| Marker | Diversity index | OTU | A13vsC13 | X13vsC13 | A14vsC14 | X14vsC14 | Manual vs. mechanical |
|--------|-----------------|-----|----------|----------|----------|----------|-----------------------|
| 16S rRNA gene | Bray–Curtis | OTUs | 0.693 | 0.249 | 0.932 | 0.045 | 0.991 |
| | | aOTUs | 0.733 | 0.223 | 0.918 | 0.058 | 0.989 |
| | Unifrac unweighted | OTUs | 0.698 | 0.298 | 0.032 | 0.029 | 0.976 |
| | | aOTUs | 0.737 | 0.407 | 0.044 | 0.018 | 0.962 |
| *gyrB* | Bray–Curtis | OTUs | 0.257 | 0.208 | 0.098 | 0.105 | 0.999 |
| | | aOTUs | 0.156 | 0.138 | 0.041 | 0.028 | 0.998 |
| | Unifrac unweighted | OTUs | 0.241 | 0.520 | 0.134 | 0.804 | 0.999 |
| | | aOTUs | 0.203 | 0.467 | 0.026 | 0.111 | 0.957 |
| ITS1 | Bray–Curtis | OTUs | **<0.001** | 0.763 | **<0.001** | 0.504 | 0.999 |
| | | aOTUs | **<0.001** | 0.777 | **<0.001** | 0.406 | 0.999 |

a threshing machine, suggesting that the harvesting method did not impact the structure of microbial assemblages (Table 1).

The impact of pathogen transmission was evaluated on microbial richness using OTU and aOTU counts as proxies for species richness. Overall, the number of bacterial OTU (Figs. S2A and S2B) and aOTU (Figs. 2A and 2B) was constant between control samples and seed harvested from A2013, A2014, X2013 and X2014. Therefore, the presence of *Xcc* 8004 and *Ab* Abra43 within the seed microbiota does not seem to alter bacterial richness. While fungal richness was also not affected by seed transmission of *Xcc* 8004, the number of fungal OTU and aOTU significantly ($P < 0.01$, as assessed by ANOVA with post hoc Tukey's HSD test) decreased for A2013 and A2014 (Fig. 2C and Fig. S2C). Hence *Ab* Abra43 seems to reduce the number of resident fungal taxa associated to radish seeds.

We next measured the effect of *Xcc* and *Ab* transmission on microbial diversity. According to inverse Simpson diversity index, bacterial $\alpha$-diversity was neither affected by *Xcc* 8004 nor *Ab* Abra43 transmission (Figs. 2D, 2E, Figs. S2D and S2E). Changes in resident bacterial assemblages between samples was further estimated by Bray–Curtis dissimilarity measure (*Bray & Curtis, 1957*) and unweighted UniFrac distance (*Lozupone & Knight, 2005*) using OTU and aOTU counts obtained with 16S rRNA gene and *gyrB* sequences. According to hierarchical clustering and ANOSIM tests, seeds harvested from inoculated plants were not significantly different from seeds collected in the control plot (Figs. S3, S4 and Table 1). This suggests that neither *Xcc* 8004 nor *Ab* Abra43 impacted the structure of seed-associated bacterial assemblages.

Regarding seed-associated fungal assemblages, no significant difference in $\alpha$-diversity was observed between control seed samples and samples contaminated with *Xcc* (Fig. 2F and Fig. S2F). However, a significant reduction of inverse Simpson diversity index was observed following invasion of seed-associated fungal assemblages by *Ab* (Fig. 2F and Fig. S2F). To gain more insight on the influence of *Ab* on seed-associated fungal assemblages, $\beta$-diversity was estimated with Bray–Curtis dissimilarity measure. A significant disturbance ($p < 0.001$) of seed-associated fungal assemblages was observed for samples harvested in

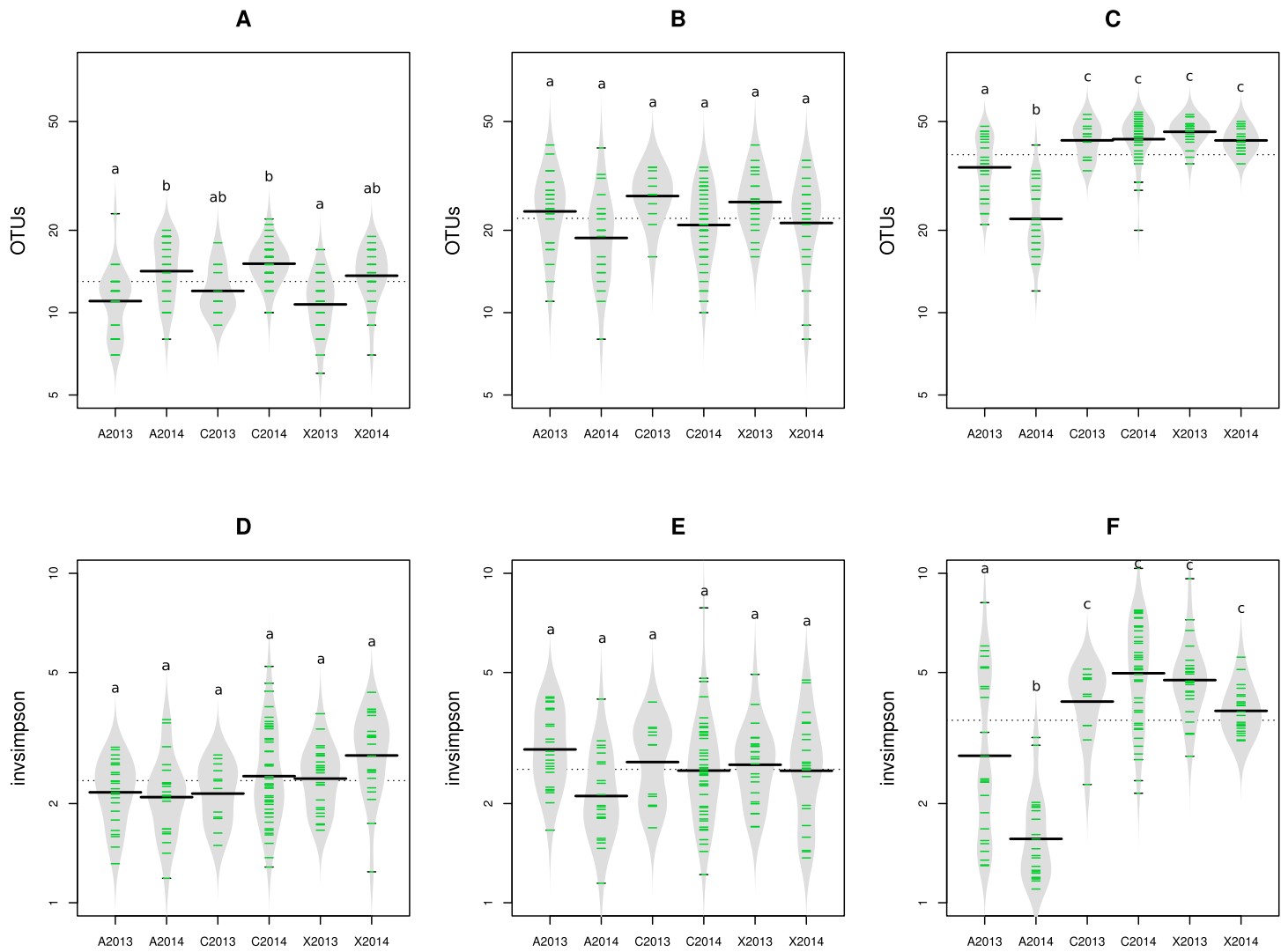

**Figure 2  Richness and diversity of seed samples observed with abundant OTUs.** Microbial richness (A–C) and diversity (D–F) were estimated with abundant OTUs obtained with 16S rRNA gene (A and D), *gyrB* (B and E) and ITS1 sequences (C and F). Richness and diversity associated to uncontaminated seeds (C2013 and C2014), seeds contaminated with *Xcc* (X2013 and X2014) and seeds contaminated with *Ab* (A2013 and A2014) were compared. Each sample is represented by a green line, while black line represents the median. The grey area represents the density of distribution. Letters a, b and c denote significant changes between conditions considered at a *P*-value ≤ 0.01 (as assessed by ANOVA with post hoc Tukey's HSD test).

A2013 and A2014 (Fig. 3 and Table 1). Indeed all these seed samples grouped together, which indicates that transmission of *Ab* Abra43 from plant to seed has a profound influence on the structure of seed-associated fungal assemblages (Fig. 3). According to canonical analysis of principal coordinates (CAP), the seed transmission of *Ab* was explaining 63% (*p* < 0.001) of the variation in fungal diversity across seed samples.

## Shift in relative abundance of microbial taxa following seed transmission of *Ab* Abra43

The taxonomic composition of seed-associated microbial assemblages was investigated in samples harvested in C2013 and C2014. According to both 16S rRNA gene and

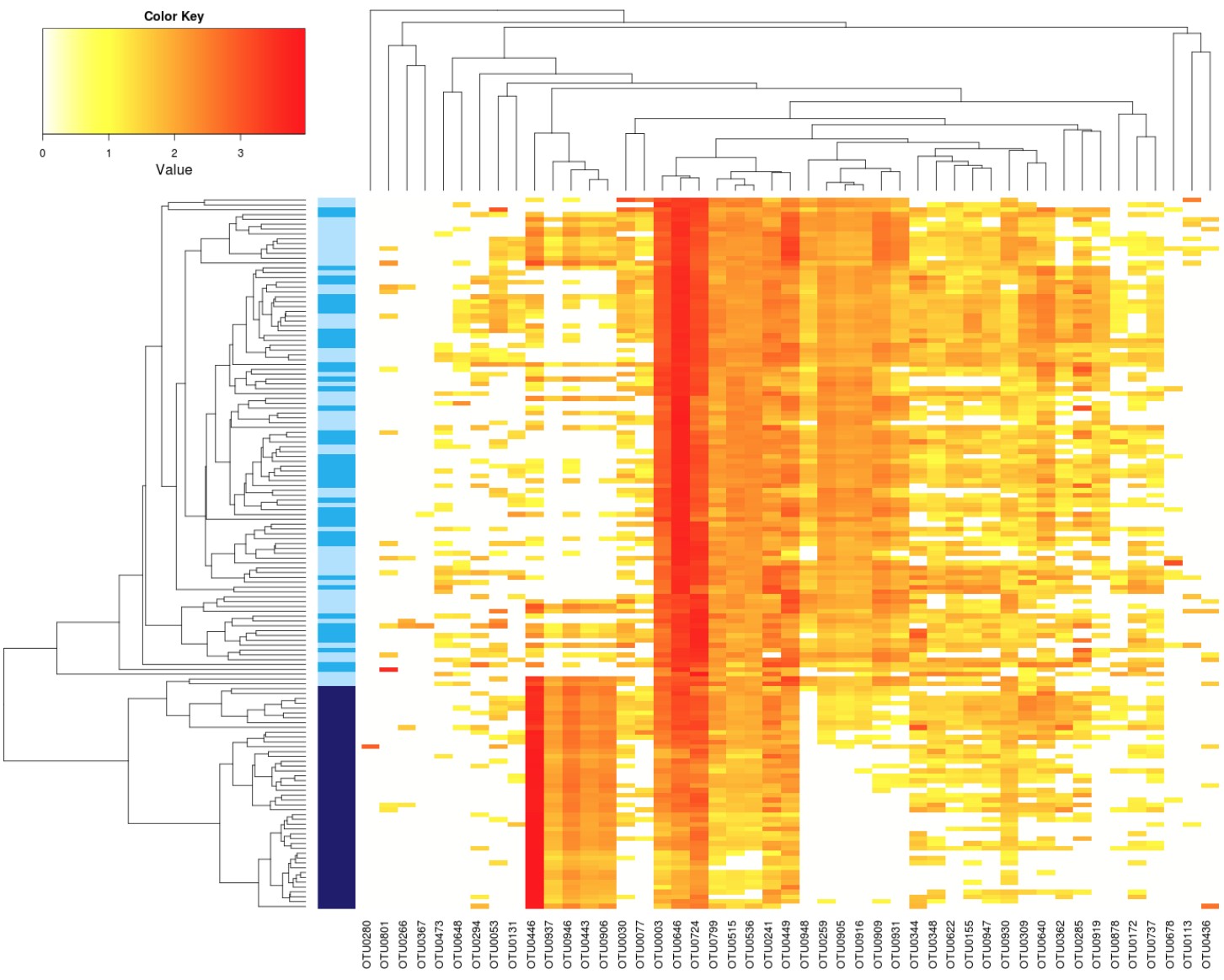

**Figure 3** **Influence of *Ab* on the structure of seed-associated fungal assemblages.** Hierarchical clustering of seed samples (*y* axis) is based on Bray–Curtis dissimilarity measure. The type of samples is represented by gradual color changes: light blue for controls, medium blue for seeds contaminated with *Xcc* and dark blue for seeds contaminated with *Ab*. Only abundant OTUs (threshold of 1% in relative abundance) are represented in the heatmap. These aOTUs are clustered by their co-occurrence patterns (*x* axis). According to analysis of similarity, a significant clustering of *Ab* seed samples was observed ($p < 0.001$).

*gyrB* sequences, bacterial aOTUs were mostly affiliated to Enterobacteriales and Pseudomonadales (Fig. S5), which confirms that taxonomic classification performed with these two molecular markers give similar results at high taxonomic rank (e.g., order level). Distribution of bacterial aOTU was then investigated across seed samples. Only 3 16S rRNA aOTUs affiliated to *Pantoea* (Otu00001) and *Pseudomonas* (Otu00002 and Otu00003) were shared across all seed samples (Fig. S3). Three *gyrB* aOTUs corresponding to *Pantoea agglomerans* (Otu00001), *Pseudomonas viridiflava* (Otu00002) and *Erwinia tasmaniensis* (Otu00003) were also conserved between all samples (Fig. S4). These aOTUs

were highly-abundant in all seed samples with an average relative abundance of 58% (*P. agglomerans*), 12% (*P. viridiflava*) and 4% (*E. tasmaniensis*) of all *gyrB* sequences. While we did not identify bacterial aOTUs specifically associated to C, A and X samples, significant changes (*p*-value ≤ 0.05 and LDA score ≥2) in relative abundance of bacterial aOTUs were observed with LEfSE (*Segata et al., 2011*). Unsurprisingly, the relative abundance of aOTUs affiliated to *Xanthomonas* (Otu0004—16S rRNA gene sequences) and *Xanthomonas campestris* (Otu00039—*gyrB* sequences) were both increased in seed samples harvested from X2013 and X2014 (Tables S3 and S4). The increase in relative abundance of these *Xanthomonas*-related aOTUs was associated with changes in relative abundance of bacterial aOTUs belonging to the Pseudomonodaceae and Enterobacteriaceae (Tables S3 and S4) and of fungal aOTUs mainly related to *Alternaria* (Table S5).

Regarding fungal assemblage composition, the seed microbiota of C2013 and C2014 samples was mainly composed of Pleosporales (Fig. S5). A significant shift (*p* < 0.01) in relative abundance of this fungal order was observed in A2013 and A2014 samples (Fig. S5). Since *Ab* belonged to the Pleosporales, this increase is probably due to the seed transmission of this pathogenic agent. At the aOTU-level 8 fungal entities were conserved in all seed samples (Fig. 3), five of them were affiliated to Pleosporales, two to Capnodiales and one to Basidiomycota. According to LEfSE analysis, 7 aOTUs were enriched in A2013 and 2014 samples (Fig. 3 and Table S5). These entities belonged to *Alternaria* and three of them are affiliated to *Alternaria* sect. *brassicicola* (OTU0443, OTU0446 and OTU0937). However, closer examination of the representative sequences of these aOTUs revealed that Abra43 belonged to OTU0446 and is the dominant phylotype of this group (96% of all reads included in OTU0446). Seed transmission of Abra43 also results in decrease in relative abundance of 61 fungal aOTUs that belonged to various fungal genus including *Alternaria*, *Cladosporium* or *Fusarium* (Table S5). In addition 7 (16S rRNA gene sequences) and 14 (*gyrB* sequences) bacterial aOTUs were also impacted by Abra43 transmission (Tables S3 and S4).

## Correlation between microbial taxa within the seed microbiota

In order to predict microbial interactions within seed-associated assemblages, we explored positive and negative associations between entities of these assemblages by generating correlations networks with SparCC (*Friedman & Alm, 2012*). Considering only inferred correlations with pseudo *p*-values ≤ 0.001, we identified 15 and 100 nodes in control samples with 16S and *gyrB* aOTUs, respectively (Fig. 4, Fig. S6 and Table S6). These nodes were sharing a total of 52 edges with 16S aOTUs and 206 edges with *gyrB* aOTUs. The other bacterial correlations networks generated with samples from A and X plots were not different from the network inferred with samples from control plots (Fig. 4, Fig. S6 and Table S6). Indeed all bacterial networks were split into multiples small modules with low connectivity between nodes. Moreover these inferred correlation networks were characterized by nodes having a maximal degree of 4–7 edges for 16S OTUs and 4–6 connections for *gyrB* aOTUs.

In comparison to bacterial networks, correlations networks inferred from seed-associated fungal assemblages were composed of more edges and less modules (Fig. 4 and Table S5),
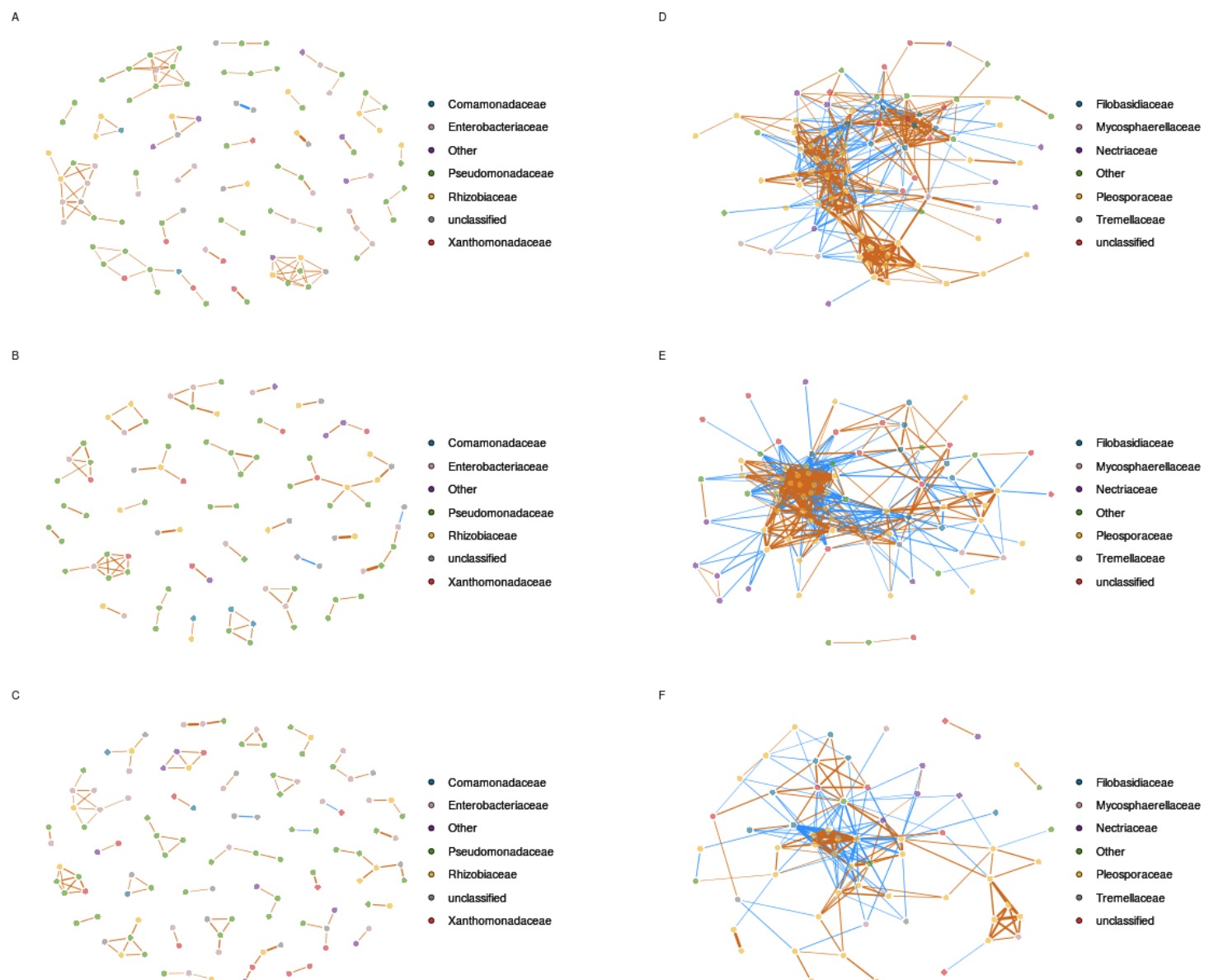

**Figure 4 Inferred correlations between aOTUs.** Correlation networks between bacterial taxa are based on *gyrB* sequences obtained in uncontaminated seeds (A), seeds contaminated with *Xcc* (B), and contaminated with *Ab* (C). Correlation networks between fungal taxa are based on ITS1 sequences obtained in uncontaminated seeds (D), seeds contaminated with *Xcc* (E), and contaminated with *Ab* (F). Correlations between aOTUs were calculated with the Sparse Correlations for Compositional data algorithm. Each node represents an aOTUs, which is colored according to its taxonomic affiliation (family-level). Edges represent correlations between the nodes they connect with blue and orange colors indicating negative and positive inferred correlation, respectively. Only correlations with pseudo *p*-value ≤ 0.001 were represented in the network using the R package qgraph.

suggesting more interactions between entities of these assemblages. Seed transmission of *Xcc* 8004 did not impact the overall structure of the inferred fungal network, since the number of nodes, edges and median number of connectivity were constant between samples from C and X plots. Moreover, the highest degree nodes of both networks were related to fungal aOTUs conserved in each seed samples such as OTU0449 (*Alternaria*

*brassicae*) for C and OTU0536 (*Alternaria* sect. *infectoria*) for X plots. On the contrary, *Ab* Abra43 strongly impacted the topology of the network with a decrease of nodes, edges and median number of connectivity (Fig. 4 and Table S6). Although OTU0449 and OTU0536 are still part of the fungal network in samples from A plots, these entities are not hubs anymore and shared 2 and 5 connections with other nodes, respectively.

Based on *gyrB* and 16S rRNA gene datasets, we were not able to detect inferred correlations between the bacterial aOTU affiliated to *Xcc* and other entities of seed-associated bacterial assemblages. Analysis of inferred correlation between *Ab* (OTU0446) and other fungal aOTUs revealed positive correlations with 7 entities that belong exclusively to *Alternaria*. Four of these aOTUs were already highlighted by hierarchical clustering (Fig. 3). In addition, 12 negative correlations between OTU0446 and other fungal entities were observed. Some of these fungal aOTUs were affiliated to *Alternaria* but also to other fungal orders such as Filobasidiales and Sporidiobolales. Interestingly the highest degree node of the correlation network inferred in samples harvested from A plot (OTU0309) was negatively associated to *Ab*, suggesting that seed transmission of this phytopathogenic fungus severely impacted resident fungal community.

## DISCUSSION

The aim of the present study was to assess the influence of two microbial invaders, namely *Xanthomonas campestris* pv. *campestris* (*Xcc*) 8004 and *A. brassicicola* (*Ab*) Abra43, on the structure of resident microbial assemblages associated with seeds. According to our indicators of bacterial (16S rRNA gene and *gyrB* sequences) and fungal (ITS1) diversity, seed transmission of the bacterial strain *Xcc* 8004 did not impact the overall composition of seed-associated microbial assemblage. In contrast, transmission of Abra43 significantly changed the structure of resident fungal assemblages without altering bacterial assemblages' composition.

Variation in response of resident microbial assemblages to invasion of *Xcc* 8004 and Abra43 could be explained by the distinct transmission pathways employed by these two micro-organisms to invade seeds, which ultimately result in differences of spatio-temporal distribution of these strains within the seed habitat. *Xcc* has been reported to invade seeds via the vascular tissue of the mother plant (*Cook, Larson & Walker, 1952*) or by flower infection during pollination (*Van Der Wolf & Van Der Zouwen, 2010*). Therefore, successful seed transmission of *Xcc* is probably strongly dependent of local resources available in the plant xylem or on flower surface. Conversely, *Ab* invades the seed via colonization of the pods and subsequent migration through the funicles (*Singh & Mathure, 2004*) and thus interacted mostly with microbial taxa associated to pods. As a consequence of these different routes of infection, *Xcc* is mostly located in the endosperm or inner integuments (*Maude, 1996*), while *Ab* is frequently isolated from the hilum of the seed coat (*Knox-Davies, 1979*). Therefore the contrasted response of seed-associated microbial assemblages to invasion by *Xcc* or *Ab* could be due to differences in microbial interactions occurring within these distinct micro-habitats. Differences in seed transmission pathways between *Xcc* and *Ab* also results in distinct timing of seed colonization, *Xcc* being associated

with earlier seed development stage. Recently, it has been hypothesized that assembly history may determine the structure of seed-associated bacterial assemblages (*Aleklett & Hart, 2013*; *Klaedtke et al., 2015*) in a similar manner than other plant-related habitat such as the phyllosphere (*Maignien et al., 2014*). Therefore, the resistance of seed-associated bacterial assemblage to *Xcc* 8004 and Abra43 invasions could be due to prior colonization of the seed by pioneer bacterial entities, which maintain community structure (*Shade et al., 2012*). Although this hypothesis has to be tested experimentally through temporal survey of microbial assemblages during the different seed development stages, it is tempting to speculate that these pioneer species may be related to bacterial taxa such as *Erwinia*, *Pseudomonas* or *Pantoea*. Indeed these taxa are highly abundant within the seeds samples collected in this study and have been frequently observed in flower (*Aleklett, Hart & Shade, 2014*) and seeds (*Johnston-Monje & Raizada, 2011*; *Links et al., 2014*; *Barret et al., 2015*; *Klaedtke et al., 2015*) of several plant species.

Alternatively, we cannot rule out the possibility that the observed stability of microbial assemblage in response to *Xcc* 8004 transmission is partly due to the low abundance of *Xcc* detected within seed samples contaminated with this bacterial strain. Although the incidence of transmission of *Xcc* on radish seeds is in accordance with previous studies (*Cook, Larson & Walker, 1952*; *Van Der Wolf & Van Der Zouwen, 2010*; *Van der Wolf, Van der Zouwen & Van der Heijden, 2013*) and reflected the natural seed transmission observed with other bacterial pathogens (*Darrasse et al., 2007*), the high prevalence of seeds not contaminated with *Xcc* in X2013 (94%) and X2014 (72%) samples together with a low *Xcc* population sizes may result in a high background signal that is likely to mask changes occurring within microbial assemblages. In comparison the high incidence and abundance of *Ab* within A2013 and A2014 seed samples allow detection of changes in microbial assemblages' structure.

In contrast to results obtained with *Xcc*, the seed transmission of Abra43 significantly impacted the structure of fungal assemblages. Perturbation of fungal assemblage following *Ab* transmission is probably explained by competition between *Ab* and other functional equivalent species for resources and spaces (*Burke et al., 2011*). Indeed the transmission of *Ab* from plant to seed is correlated with a decrease in relative abundance of closely related entities that belong to *Alternaria brassicae*, *Alternaria*. sect. *infectoria* and *Alternaria*. sect. *alternate* (*Woudenberg et al., 2013*). As these fungal aOTUs represented hubs of inferred fungal correlation network in native condition, the potential competition of these entities with *Ab* result in a drastic shift in the structure of seed-associated fungal assemblage. Among the observed changes, we observed multiple co-occurrences between *Ab* and other fungal entities related to *Alternaria* sect. *brassicicola*. Whether these entities either interact positively or are selected in similar ways by the environment remained to be determined.

The results of this work provide a first glimpse into the response of the seed microbiota following seed transmission of two phytopathogenic microorganisms. Future metagenomics analysis of seed-associated microbial assemblages will be useful to assess the relationship between assemblage structure and function (*Vayssier-Taussat et al., 2014*). This research might lead to the development of biocontrol strategies based on the potential of seed-associated microbial community.

## ACKNOWLEDGEMENTS

The authors wish to thank Julie Gombert and Vincent Odeau (FNAMS) for their help with all the field experiments and Muriel Bahut and Laurence Hibrand-Saint Oyant from the platform ANAN of SFR Quasav for their help on the MiSeq experiments.

### Funding

This research was supported in parts by grants awarded by the Region des Pays de la Loire (Qualisem, 2009 05369 and metaSEED, 2013 10080) and the European Commission (TESTA, FP7-KBBE-2012-6, 311875). The funders had no role in study design, data collection and analysis, decision to publish, or preparation of the manuscript.

### Grant Disclosures

The following grant information was disclosed by the authors:
Region des Pays de la Loire: Qualisem, 2009 05369, metaSEED, 2013 10080.
European Commission: TESTA, FP7-KBBE-2012-6, 311875.

### Competing Interests

The authors declare there are no competing interests.

### Author Contributions

- Samir Rezki performed the experiments, analyzed the data, wrote the paper, prepared figures and/or tables.
- Claire Campion and Emmanuelle Laurent conceived and designed the experiments, performed the experiments, reviewed drafts of the paper.
- Beatrice Iacomi-Vasilescu performed the experiments, reviewed drafts of the paper.
- Anne Preveaux, Youness Toualbia and Sophie Bonneau performed the experiments, contributed reagents/materials/analysis tools, reviewed drafts of the paper.
- Martial Briand analyzed the data, reviewed drafts of the paper.
- Gilles Hunault analyzed the data, prepared figures and/or tables, reviewed drafts of the paper.
- Philippe Simoneau conceived and designed the experiments, wrote the paper.
- Marie-Agnès Jacques conceived and designed the experiments, wrote the paper, prepared figures and/or tables.
- Matthieu Barret conceived and designed the experiments, performed the experiments, analyzed the data, wrote the paper, prepared figures and/or tables.

### DNA Deposition

The following information was supplied regarding the deposition of DNA sequences:
All sequences have been deposited in the ENA database under the accession number PRJEB9588.

## Data Availability

Raw data have been deposited in the ENA database
http://www.ebi.ac.uk/ena/data/view/PRJEB9588.

## Supplemental Information

Supplemental information for this article can be found online at http://dx.doi.org/10.7717/peerj.1923#supplemental-information.

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
