# Peer review of "Differences in stability of seed-associated microbial assemblages in response to invasion by phytopathogenic microorganisms"

_PeerJ, doi:10.7717/peerj.1923_

## Round 0.1 · original submission · Minor Revisions

I found your work of great interest and merits publication after minor modifications, because it provides interesting and new information. I agree with reviewer 1 concerning the need of clarification throughout the paper that you particularly studied the seed surface associated microorganisms, since you did extract the DNA from seed washes not total seed tissue. So, please avoid the term seed microbiota as it implicates that you studied also the endophytic seed-associated microorganisms and replace it by seed surface associated or seed epiphytic microbiota.

Also you indicated that unclassified sequences or sequences belonging to Eukaryota, Archaea, chloroplasts or mitochondria were discarded. Can you provide the percentage of sequences of each type that you discharged? This information can be very valuable for researchers that want to use same primers with total plant DNA tissue.

Reviewer 1 ·

Basic reporting

No comments

Experimental design

It is not clear at all to me whether the authors studied the microbiota on the surface of seeds or both the surface colonizing microorganisms and the microorganisms colonizing the interior of seeds. The authors are very vague in their wording in this context throughout the paper. E.g. line 161-163: Does the pellet used for DNA isolation contain the imbibed seeds or whatever was washed of from the surface of the seeds? Later the authors use the primers 515F/806R for amplification of bacterial 16S rRNA gene fragments. This implicates that the authors did not analyze the endophytic microorganisms in the seeds as this would have resulted most probably in massive co-amplification of plant plastid small rRNA gene fragments. On the other hand the authors talk about the seed microbiota in Results and Discussion.
Please clarify throughout the paper and in case you studied the seed surface associated microorganisms please avoid the term seed microbiota as it implicates that you studied also the endophytic portion of the seed-associated microorganisms. Seed surface associated or seed epiphytic would be more appropriate in this case .

Validity of the findings

No comments.

Additional comments

Overall, the manuscript is very well written, the experimental set-up sound and the results presented clearly. The only weak point in this manuscript is that it is not clear whether the authors studied also the endophytes of the seeds or the seed-surface bacteria and fungi only. This needs to be clarified throughout the paper before publication.

Reviewer 2 ·

Basic reporting

No comments

Experimental design

No comments

Validity of the findings

No comments

Additional comments

The manuscript submitted by Rezki et al reports the results of an original study that was designed to evaluate the role of two seed transmitted pathogens (Xanthomonas campestris pv. campestris and Alternaria brassicicola) in the Raphanus sativus seed bacterial assemblages. The seed microbiota has been poorly investigated and the authors used the Raphanus as a model to contribute to fill the gap about this relevant topic.
The experiments are well described although some more details could be added in the Materials and methods section. The results and conclusions of the work are sound but some of the analysis could require additional explanations. They show the potential of some seed transmitted pathogens in modifying seed microbiota.
The manuscript could be improved following the suggestions indicated below:

Specific suggestions:
- Line 30. Indicate that the study was performed in Raphanus sativus seeds.

- Line 95. Add genetic before structure and Raphanus before seed.

- Line 114. Were the seeds used in 2013 and 2014 previously analysed for detecting the presence of Xanthomonas campestris and Alternaria brassicicola?. Indicate this information in the text.

- Line 116. There is no data about the behaviour in seed and in planta of the mutant of X. campestris pv. campestris utilised in the inoculations, compared to a wild strain. It is important to know if the authors found differences among them in growth, colonisation, survival or pathogenicity.

- Line 132. Indicate the conditions of the assay after inoculation and if they were favourable for the multiplication of the bacterial and the fungal strains inoculated.

- Line 133. Indicate the time and conditions of conservation of the seeds between the time of harvesting the seeds and the DNA extraction.

- Line 189. Indicate briefly in the text the origin of the 148 DNA samples.

- Line 248. It should be interesting to add comments about the symptoms of X. campestris pv. campestris and A. brassicicola observed in the plants inoculated in 2013 and 2014 and the % of plants showing typical symptoms of both pathogens.

- Discussion. As some of the statistical analyses performed can not be well understood by the large audience of PeerJ, the readers will appreciate more conclusions about the biological relevance of the statistically significant differences found among the data analysed, in the context of this work.

In conclusion, the submitted manuscript merits publication after minor modifications, because it provides interesting and new information using the most appropriated and updated tools and analyses.

---

## Round 0.2 · accepted · Accept

I think that although reviewer 1 still have some doubts concerning the validity of your technique for being extrapolated to endophitic microorganisms the term you used throughout the manuscript 'microbial assemblages associated to seeds' is valid to me.

When in production please try to clarify sentence in line 63-64 and also change sentence in line 309-312 to … “According to ANOSIM tests, the structure of seed-associated microbial assemblages on seed samples harvested manually are not significantly different ….

Reviewer 1 ·

Basic reporting

line 63-64: This sentence is somehow misleading. One gets the Impression that you talk about floral transmission to the seed but the study of Spinelli et al. (2005) is on flower colonization and seeds have not been analyzed. Please clarify in the text.

Experimental design

I do not agree with the authors that a recommendation for the detection of seed born pathogens given by the International Seed Testing Association is enough evidence that this procedure allows for the analysis of endophyic microbiomes. I actually think that the ISTA procedure might cause a severe bias in the analysis.
If the authors prefer to say that they analyzed the seed microbiome, they certainly should provide an experimental proof, i.e. compare community sequencing data obtained with their approach with community data obtained by DNA isolation of total seeds.

Validity of the findings

The data are robust and clearly presented.